# Anisotropic Characterizations of Electrospun PAN Nanofiber Mats Using Design of Experiments

**DOI:** 10.3390/nano10112273

**Published:** 2020-11-17

**Authors:** Blesson Isaac, Robert M. Taylor, Kenneth Reifsnider

**Affiliations:** 1Chemical and Radiation Measurement Department, Energy Environment Science and Technology, Idaho National Laboratory, Idaho Falls, ID 83415, USA; 2Department of Mechanical and Aerospace Engineering, The University of Texas at Arlington, Arlington, TX 76019, USA; taylorrm@uta.edu (R.M.T.); kenneth.reifsnider@uta.edu (K.R.)

**Keywords:** design of experiment, electrospun fibers, tensile, dielectric

## Abstract

This paper deals with the dielectric and mechanical characterizations of polyacrylonitrile (PAN)-aligned electrospun nanofiber mats. A two factor three level full factorial experiment is conducted to understand the effect of various parameters on dielectric and mechanical responses. These responses are recorded against randomly oriented and aligned nanofiber mats. Improved properties of electrospun mats have applications in the field of energy storage and nanocomposite reinforcement. Dielectric and mechanical characterizations of PAN mats are vital, as the aligned electrospun mats were found to be useful in advanced energy and mechanical reinforcement applications. Therefore, it is paramount to understand the effects of system parameters to these properties. The design of experiment (DoE) includes two factors and three level full factorial experiments with concentrations of PAN solutions at 8 wt.%, 9 wt.%, and 10 wt.%, and speed of the rotating mandrel (collector) at 3 volt (V), 4 V, and 5 V inputs. The electric field intensity used in the experiment is 1 kV/cm. DoE is conducted to understand the nonlinear interactions of parameters to these responses. The dielectric and mechanical characterizations of 8 wt.%, 9 wt.%, and 10 wt.% with different speeds for the original and improved systems are discussed. It was observed that at 9 wt.% and at all mandrel speeds, the dielectric and tensile properties are optimum.

## 1. Introduction

Taylor (1964) showed that liquid disintegrates due to instability when subjected to a uniform electric field [1,2]. Coulomb’s force is thought to be the main cause for instability. Carbon nanofibers can be fabricated with conventional methods like catalytic synthesis, vapor grown, and arc discharge [3]. However, electrospinning is a cost effective, versatile, top down engineering approach. Electrospinning is considered as a powerful technology for the production of nanofibers with diversified architecture and properties [4,5]. It has applications in the fields of energy and environment [6] and biomedical [7]. Polymeric nanofiber non-woven mats produced by electrospinning have high specific areas (high surface area to volume ratio) and porous structures that make them suitable for a wide range of applications [8]. DoE is conducted to identify the optimization of processing parameters and materials properties of electrospun fibers [9,10,11,12,13]. PAN is a widely available precursor for the manufacturing of carbon nanofibers. Arshad et al. (2011) showed that the maximum ultimate strength for PAN fibers occurred at 1 kV/cm [14]. In addition, Mei et al. (2013) reported that maintaining the electric field intensity at 1 kV/cm produced optimal alignment [15]. Popkov et al. (2013) showed that different concentrations from 8 to 11 wt.% polymer solutions have PAN nanofibers with extraordinary strength, modulus, and toughness [16]. Jalili et al. (2006) showed that the best alignment for PAN/dimethyl formamide (DMF) occurs for 10–15% using parallel plates [17]. However, Khan et al. (2015) showed that this value was 6–12 wt.% concentration with increasing fiber diameter from 6 wt.% [18]. Uniform PAN nanofibers were obtained at concentrations of 8 and 10 wt.% under the applied voltage of 10–20 kV [19]. According to Gu et al. (2005), applied voltage had no significant influence on fiber diameter, whereas the concentration of the solution impacted the diameter of fibers. The average fiber diameter increased with polymer concentration according to a quadratic relationship. Alignment maximizes modulus and strength and make the composite anisotropic [20]. Fiber diameter is a major parameter for determining properties of electrospun fibers [21]. With solvent and polymer solute, there are three types of spinning: wet, dry, and gel. An additional technique, melt spinning, uses no solvent. For that method, the polymer precursor can be prepared by melting the polymer at an elevated temperature between 250 and 400 °C in the absence of a solvent.

PAN-based nanofibers are widely used for carbon fiber production [22]. PAN is soluble in polar solvents like DMF, dimethyl sulfone (DMS), dimethyl sulfoxide (DMSO), and dimethyl acetamide (DMAc) [23]. Among organic solvents, DMF and DMSO are known to be good solvents of PAN [24]. Among the various organic and inorganic solvents, DMF offered the best properties. Optimization of PAN precursor fibers results in enhanced performance for use in aerospace applications [18]. Edmondson et al. (2012) demonstrated the capability of their system using PVDF as a model polymer, which has favorable piezo-, pyro-, and ferro-electric properties, and showed that aligned PVDF fibers can provide for applications in actuators, transistors, textiles, and composites [24,25]. Electrospinning is also used to increase the specific area, alignment of fibers, and β-phase content for polymorphic materials.

Fiber alignment can improve the piezoelectric effect of voltage generation [24]. The β-phase increment improves the piezoelectric effect of nanofibers. The β-phase increment is due to shearing force through the needle, columbic force between the collector and the needle, and mechanical force due to elongation of fibers on the rotating drum. Depending on the applications, material properties generated or enhanced through electrospinning are exploited using appropriate polymer selections. In our scope of study, two applications are important, namely specific area for composite reinforcement (PAN is an example for the polymer) and β-phase for energy generation of electrode cells (PVDF is an example for polymer). 

## 2. Materials and Methods

PAN powder purchased from Sigma Aldrich Inc, Milwaukee, US with product number 181315 was mixed with DMF bought from Sigma Aldrich Inc, St. Louis, US with product number D158550. The molecular weight of PAN is 150,000 Dalton (Da). They were mixed in conical glassware that had a stopper. Eight wt.% of solute in the solvent were stirred and heated at 300 rpm and 80 °C for 4 h until a homogeneous yellow colored solution was formed. For 9 and 10 wt.%, the stirring could change. The viscosity of solutions were measured using an SV-10 viscometer, and the viscosity of 10 wt.% solution was 28.4 Poise, or 2.84 Pa –s, at 20 °C. The syringe and needle were purchased from Becton and Dickinson. The syringe was 5 mL and the needle was 22 G × 1 1/2 (0.7 mm × 40 mm) in dimensions. Figure 1 shows the stirrer and PAN solution after heating and stirring. The solution was pumped through a programmable syringe pump. The input voltages and the speeds of the mandrel were proportional: 5 V, 4 V, and 3 V input voltages and 3100 rpm, 2400 rpm, and 1700 rpm, respectively.

## 3. Systems and Modifications

The original system had four basic components: high voltage power supply, syringe, needle, and a collector. They were enclosed in a closed chamber. The collector was a rotating type of horizontal mandrel. The improved system had an additional negative electric potential. The syringe and needle moved at a controlled rate along the horizontal direction of the mandrel. The original and improved systems were compared through morphological analysis.

### 3.1. Working Principle

The schematics of the improved system are shown in Figure 2. The improved system had the capability of reducing the repulsion of fibers as they were formed on the mandrel. The deposition of fibers was controlled by the linear movement of the syringe and needle. With the improved system, the fibers formed were aligned and uniform. The improved system supplied −6 kV in addition to the 15 kV from the high power voltage supply to the needle. The distance from needle tip to mesh tip was 20 cm. Thus, the electric field intensity was 23 KV/20 cm = 1.2 kV/cm. The volume dispensed was 1 mL. The flowchart for the electrospinning improved system is shown in Figure 3. The computer interface enabled precise movement of the linear stage connected to the programmable syringe pump and syringe needle using a bolted base plate. The needle moved accordingly along the horizontal direction of the mandrel. The syringe pump was mounted with the syringe and needle. The solution in the syringe was supplied at a controlled rate of 1 mL/h. The positive electrode and the grounded mandrel brought an electrostatic potential to form fibers on the mandrel. The negative electrode and mesh provided additional potential that pulled the fibers in the perpendicular direction of the horizontal mandrel, reducing the repulsion between the fibers.

A pictorial representation of the improved system is shown in Figure 4. Aligned fibers formed on the mesh are shown in Figure 5. These fibers were formed on the negative electrode, showing that the improvement system worked. The width of the aligned fibers was 1 inch. The difference in alignment using the improved electrospinning system and original electrospinning system is shown in Figure 6. Both the original and improved systems were kept at a 20 cm needle tip to collector distance. Both were kept at an electric field intensity of 1.2 kV/cm.

### 3.2. Morphological Analysis

The scanning electron microscope (SEM) images of the electro spun mats magnified at 20 µm for the original and improved systems with angle measurements are shown in Figure 6. The SEM images were taken using an SEM HR S-4800 device. The angle measurements were provided by ImageJ open source imaging software. There were 110 fibers, each taken to measure the angular orientation for the original and improved systems. The standard deviation in the angular orientation of all the fibers of the improved system was 53° compared to that of the original system, which was 58°. There was slightly lower variability in alignment for the improved system. Figure 7 shows that the variability of mass along the mats was reduced in the improved system. For the improved system, the coefficient of variation of mass was 5% compared to the coefficient of the variation of the original system, which was 8%. The standard deviation in mass distribution of specimens in the improved system was 1.5, and that of the original system was 2.5. There was much lower variability in mass distribution for the improved system.

## 4. Characterization for Dielectric and Tensile Tests

Various tests conducted for DoE were tensile and dielectric. The DoE test matrix for original and improved systems is shown below in Table 1.

The two factors used were speed (A) and concentration (B). Table 2 shows the three levels used for each factor. The number of samples taken for mechanical and dielectric tests were 54 each.

### 4.1. Dielectric Test

A dielectric test was conducted using a 0.5 inch × 0.5 inch electrode in a Faraday cage using a broadband dielectric spectrometer (BbDS), which uses the frequency range between 1 MHz and 0.1 Hz. An ASTM D-150 was used for this dielectric test to be conducted. The test configuration and schematic of dielectric test are shown below in Figure 8. The dielectric constants increased above 1 kHz. The interfacial polarization came into play below the frequency of 1 kHz, as the charges were accumulated at lower frequencies between the pores and fibers [8]. Values were taken only at 1 kHz for this DoE study.

The relative permittivity of nanofiber mat [26] is calculated as follows:$$\varepsilon_{r}=C\times \frac{l}{A\times \varepsilon_{o}}$$
where $\varepsilon_{r}$ is the relative dielectric constant, C is the capacitance, A represents area of plates, l is the thickness of the film, and $\varepsilon_{o}$ is the relative permittivity of the vacuum.

$\varepsilon_{r}$ is a function of porosity and density [26].
$$P\;(\%)=(1-\frac{\rho_{M}}{\rho_{P}})\times 100$$
where P is the porosity, $\rho_{M}$ is the membrane density, and $\rho_{P}$ is the polymer density.

Normalized dielectric constant in kg^−1^m,
$$\frac{\varepsilon_{r}}{\rho \times A}=\frac{\varepsilon_{r}}{M/l}$$
where M is the mass of the specimen in kg, l is the thickness of the film in m, ρ is the density of the mat in kgm^−3^, and A is the surface area of the mat in m^2^.

### 4.2. Tensile Test

The ASTM D882-12 standard test method was followed for tensile tests. From a practical perspective, porosity and thickness can be difficult to accurately measure. Mass is used instead of thickness [27]. For the improved system, the tensile tests were taken from specimen with sizes 2 inch × 1/2 inch length and width, respectively. The test configuration and schematic representation of the tensile specimen are shown in Figure 9.

Specific strength of a nanofiber mat [28] $\sigma_{sp}$,
$$\frac{F}{\rho \times A}=\frac{F}{M/L}$$
where $\sigma_{sp}$ is the specific strength of the tensile specimen in N kg^−1^m, ρ is the fiber density of fiber in kgm^−3^, A is the cross sectional area of the specimen in m^2^, M is the mass of nanofiber mat in kg, and L is the gauge length of the specimen in m. In addition,
$$\sigma_{sp}=\frac{\sigma }{\rho \left(1-P\right)}\;N\;kg^{-1}m$$
where σ is the tensile strength of nanofiber mat in Nm^−2^ and P is the porosity.
$$P\;(\%)=(1-\frac{\rho_{M}}{\rho_{P}})\times 100$$
where P is the porosity, $\rho_{M}$ is the membrane density, and $\rho_{P}$ is the polymer density.

## 5. Results and Discussion

The results of dielectric and mechanical tests for DoE are given below. The dielectric and tensile test responses are discussed here.

### 5.1. Dielectric—Main Effects and Interactions

Figure 10 shows the main effects, standard deviation, and interaction between the factors for dielectric data from the original system. For Factor A, the difference in main effects was 3.8 kg^−1^m from A− to A0. The lowest value was 88.4% of the highest value. The coefficients of variations were 27.2% and 28.3%, respectively. The difference in the main effects was 8.9 kg^−1^m from B− to B+. The lowest value was 72.6% of the highest value. For Factor B, the coefficients of variation were 30.6% and 34.2% at B− and B+, respectively. At the highest concentration (10 wt.%), 5 V corresponded to the highest value of the normalized dielectric constant. The interaction was more evident at 9 wt.%, and 3 V contributed to the highest value of a normalized dielectric constant.

Figure 11 shows the main effects, standard deviation, and interactions between the factors for dielectric data from the improved system. For Factor A, the difference in the main effects was 6.2 kg^−1^m from A− to A+. The lowest value was 72.3% of the highest value. For Factor B, the difference in the main effects was 5.9 kg^−1^m from B− to B0. The lowest value was 73% of the highest value. The coefficients of the variation were 56% and 26% at B− and B0, respectively. At the concentration of 9 wt.%, 3 V corresponded to the highest value of the normalized dielectric constant; the next higher value corresponded to 4 V.

### 5.2. Tensile—Main Effects and Interactions

Figure 12 shows the main effects, standard deviation, and interactions between the factors for tensile test data from the original system. For Factor A, the difference in main effects was 80.2 × 10^2^ N kg^−1^m from A+ to A0, and the lowest value was 71.4% of the highest value. The coefficients of variation were 50.3% and 36.5% at A+ and A0, respectively. For Factor B, there was a difference of 37.7 × 10^2^ N kg^−1^m from B− to B+. The lowest value was 85% of the highest value. The coefficients of variation were 50% and 21.4% at B− and B+, respectively. At the lowest concentration (8 wt.%), 5 V corresponded to the highest specific strength. At 9 wt.%, 3 V and 5 V corresponded to the next higher specific strengths.

Figure 13 shows the main effects, standard deviation, and interactions between the factors for tensile test data from the improved system. The specific tensile strength data from the improved system showed that there was a difference of 55.1 × 10^2^ N kg^−1^m in the main effects from A+ to A−. The lowest value was 79.7% of the highest value. For Factor A, the coefficients of variation were 20.5% and 27% at A+ and A−, respectively. For Factor B, the difference in main effects was 40.7 × 10^2^ N kg^−1^m from B+ to B−. The lowest value was 84% of the highest value. For Factor B, the coefficients of variation were 23.5% and 24.5% at B+ and B−, respectively. At the lowest concentration (8 wt.%), 5 V corresponded to the highest specific strength. At 9 wt.%, 5 V corresponded to the highest specific strength, 4 V corresponded to the higher value, and 3 V corresponded to the lowest value. 

Figure 14 shows that specific tensile strengths were almost uniformly distributed at 8 wt.% and at all speeds of mandrel for the improved system when compared to that of the original system, because aligned fibers were formed in the improved system, whereas randomly oriented fibers were formed in the original system. Mats produced from the improved system are useful in various applications, such as reinforcement and energy storage. All specimens were taken from two different mats fabricated under the same conditions. For example, six specimens were taken from two different mats fabricated at a 3 V speed and 8 wt.% concentration. As shown below, the 5 V speed nanofiber mats had an average tensile strength higher than that of the 4 V speed nanofiber mats. The average specific strength of 5 V and 8 wt.% specimens was 24.8 × 10^3^ N kg^−1^ m, and the average specific strength of 4 V and 8 wt.% was 20.6 × 10^3^ N kg^−1^ m. This indicates that mats with the combined effects of aligned fibers and uniform distribution of mass had better mechanical strength. The improved system brought about mats with slightly better alignment of fibers and more uniform mass distribution compared to that of the original system.

Figure 15 shows the XRD analysis of mats from the original and improved systems at 8 wt.%. Crystallinity was determined as the ratio of the peaks to the total area of the curve. Thinner fibers results in higher crystallinity [16,29]. Though the peak in the improved system was smaller, the percentage of crystallinity was higher in the improved system. This shows that the mats from the improved system had higher tensile strength. It is also noticeable that planes were distorted in the improved system, and only one plane was predominant. This shows that the improved system distorted other planes by molecular orientation due to alignment.

Figure 16 shows 4 V and 5 V speed mats for 8 wt.% of the original system, and the angular orientation was taken for 110 fibers each. The fiber alignment was slightly better at 5 V speed than at 4 V speed. The standard deviation of angular orientation of 4 V-8 wt.% was 59, whereas that of 5 V–8 wt.% was 53. The fiber direction side of the mat coincided with the vertical axis of the SEM HR S-4800, as shown in Figure 17. The alignment of fibers was measured along the fiber direction.

### 5.3. Linear Regression Model

A model that might describe an empirical type relating the tensile strength or dielectric constant of a polymer to the speed and concentration of an electrospinning rotating mandrel and solution, respectively, is
$$y=\beta_{0}+\beta_{1}x_{1}+\beta_{2}x_{2}+\beta_{3}x_{3}+\beta_{4}x_{4}+\beta_{5}x_{5}+\varepsilon$$
where y represents the tensile strength or dielectric constant, $x_{1}$ represents the speed, and $x_{2}$ represents the concentration. If we let $x_{3}=x_{1}^{2}$, $x_{4}=x_{2}^{2}$, and $x_{5}=x_{1}x_{2},$ the above equation becomes a multiple linear regression model [30]. 

Applying the method of least squares, the sum of the squares of the errors, $\varepsilon_{i}$, is minimized. The least squares function is
$$L=\overset{n}{\underset{i=1}{\sum }}\varepsilon_{i}^{2}$$

The function L is to be minimized with respect to regression coefficients $\beta_{0},$
$\beta_{1}$, $\beta_{2,}$…, $\beta_{k}$.

The least squares estimators must satisfy
$$\frac{\partial L}{\partial \beta_{0}}=0\;\mathrm{and}\;\frac{\partial L}{\partial \beta_{j}}=0\;j=1,\;2,\;3,\ldots ,\;k.$$

The least squares normal equations may be written in matrix notation as
y=Xβ+ε

The fitted regression model is
$$\hat{y}=X\;\hat{\beta }\;\hat{\beta }={\left(X^{\prime }X\right)}^{-1}X^{\prime }y$$
where $\hat{\beta }$ is the least squares estimators of the regression coefficients, β, and residual
$$e=y-\hat{y}$$

#### 5.3.1. Dielectric—The Original System

The response surface equation [9,31,32] of the normalized dielectric constant of the original system is
$$Y_{1}=5.35\times 10^{2}-0.31\times 10^{2}\times A-1.00\times 10^{2}\times B+0.02\times 10^{2}\times A^{2}+0.05\times 10^{2}\times B^{2}+0.01\times 10^{2}\times AB\;{\mathrm{kg}}^{-1}\;m$$
where $Y_{1}$ is the normalized dielectric constant of the original system in kg^−1^m, and A = A/V and B= B/wt.% are speed per voltage and concentration per wt.%, respectively.

Figure 18 shows the fitted model of the normalized dielectric constant of mats from the original system. 

The goodness of fit is shown in Figure 19. The coefficient of determination is 86.96%.

#### 5.3.2. Dielectric—The Improved System

In addition, for the improved system, the fitted model of the normalized dielectric constant is shown in Figure 20,
$$Y_{1}^{\prime }=-4.7\times 10^{2}-0.12\times 10^{2}\times A+1.16\times 10^{2}\times B+0.02\times 10^{2}\times A^{2}-0.06\times 10^{2}\times B^{2}-0.01\times 10^{2}\times AB\;{\mathrm{kg}}^{-1}\;m$$

The goodness of fit is shown below in Figure 21. The coefficient of determination is 68.43%.

#### 5.3.3. Tensile—The Original System

The response surface regression equation of the specific tensile strength is
$$Y_{2}=-2.06\times 10^{4}+1.84\times 10^{4}\times A+0.089\times 10^{4}\times B+0.42\times 10^{4}\times A^{2}+0.10\times 10^{4}\times B^{2}-0.54\times 10^{4}\times AB\;N\;kg^{-1}\;m$$
where $Y_{2}$ is the specific tensile strength of original system, and A = A/V and B = B/wt.% are speed per voltage and concentration per wt.%, respectively. Figure 22 shows the fitted model of the specific strength mats from the original system. 

The goodness of fit is shown below in Figure 23. The coefficient of determination is 71.91%.

#### 5.3.4. Tensile—The Improved System

Similarly, for the improved system as shown in Figure 24,
$$Y_{2}^{\prime }=-1.4\times 10^{5}+0.03\times 10^{5}\times A+0.31\times 10^{5}\times B+0.02\times 10^{5}\times A^{2}-0.01\times 10^{5}\times B^{2}-0.02\times 10^{5}\times AB\;{N\;\mathrm{kg}}^{-1}\;m$$

The goodness of fit is shown below in Figure 25. The coefficient of determination is 80.96%.

The improvements in specific tensile strengths for improved system at different speeds and concentrations are given in Table 3. The decreases at lower concentrations were due to less alignment and randomness of fibers because of lower electrostatic potential. The results show that for the improved system, at 5 V speed and at 10 wt.% there was an increase in the specific tensile strength from 20.9 × 10^3^ to 26.4 × 10^3^ N kg^−1^m. At 9 wt.%, there was an increase from 23.8 × 10^3^ to 30.3 × 10^3^ N kg^−1^m, and at 8 wt.% there was a decrease from 39.4 × 10^3^ to 24.8 × 10^3^ N kg^−1^m.

As shown in Table 4, for normalized dielectric constant and for 5V speed, at 10 wt.% there was a decrease from 33.9 to 19.6 kg^−1^m, at 9 wt.% there was a decrease from 24.4 to 23.2 kg^−1^m, and at 8 wt.% there was a decrease from 30.6 to 22.4 kg^−1^m. 

Similarly, the dielectric constants for speeds and concentrations are given in Table 4.

The decreases in dielectric constants were due to the alignment of fibers. The charges were not accumulated, or, in other words, the capacitance of the mat was reduced due to alignment. The effect of randomness of fibers came into play at 9 wt.%, and the alignment effect was overtaken by less porous specimens, resulting in increased dielectric constant.

The Pareto front for the original and improved systems is shown in Figure 26, and it indicates that for the original system, as the tensile strength increased from 21 to 29 kN kg^−1^m, the dielectric constant decreased from 33 to 26 kg^−1^m. For the improved system, as the tensile strength increased from 23.2 to 24.8 kN kg^−1^m, the dielectric constant decreased from 34.4 to 30.8 kg^−1^m.

## 6. Conclusions

A combined effect of aligned fibers with uniform distribution of mass is proven for nanofiber mats to have better mechanical and dielectric strength. The improved system shows higher tensile strength and a lower dielectric property at 10 wt.% concentration for all mandrel speeds. However, the increment in tensile strength was prominent at 4 V for 9 wt.% and 8 wt.% concentrations. The change in slope of tensile strength and dielectric was found at 9 wt.% for all different speeds. For composite applications, high tensile strength is preferred. The dielectric property is related to energy applications. Using the improved system, the dielectric property can be tailored for energy applications. Therefore, the PAN mats produced from the improved system are useful in nanocomposite reinforcement applications and controlled energy applications. The DoE study shows the extensive outlook of speed and concentration and their interactions when analyzing the dielectric and tensile outputs. 

## Figures and Tables

**Figure 1 nanomaterials-10-02273-f001:**
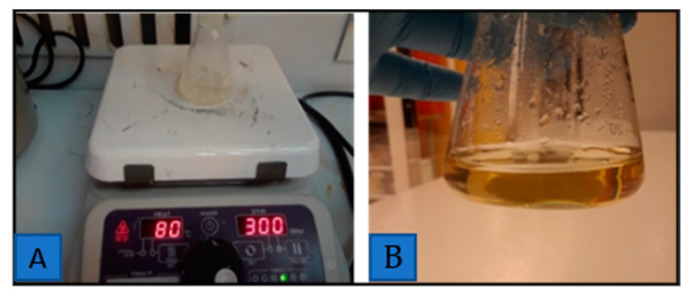
(**A**) Stirrer and heater and (**B**) yellowish solution.

**Figure 2 nanomaterials-10-02273-f002:**
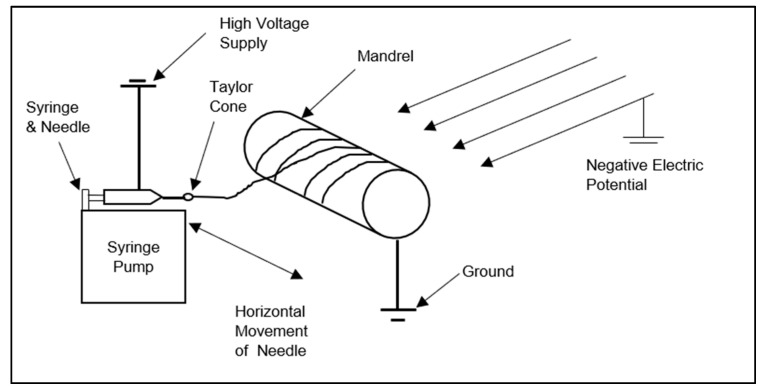
Schematics of the improved system.

**Figure 3 nanomaterials-10-02273-f003:**
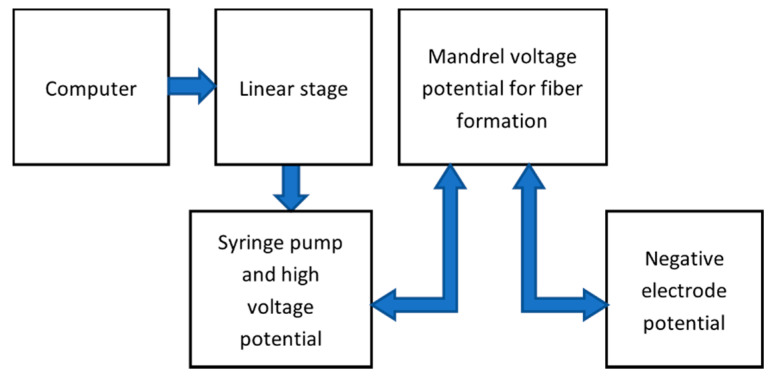
Flow chart of the improved system.

**Figure 4 nanomaterials-10-02273-f004:**
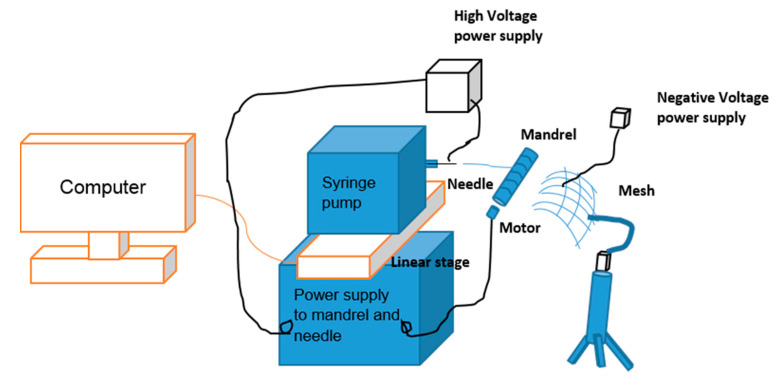
Pictorial representation of the improved system.

**Figure 5 nanomaterials-10-02273-f005:**
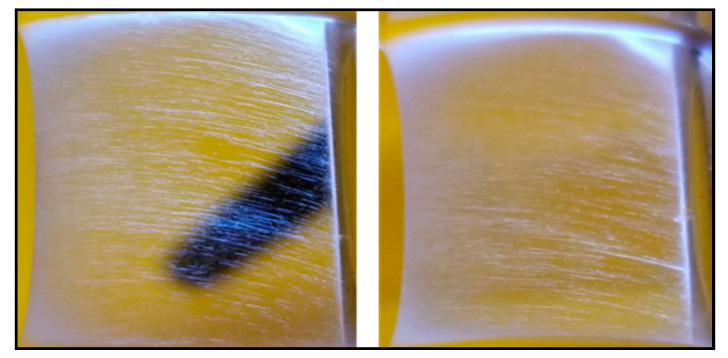
Aligned fibers formed in the mesh of the improved system (shadow is alligator clip).

**Figure 6 nanomaterials-10-02273-f006:**
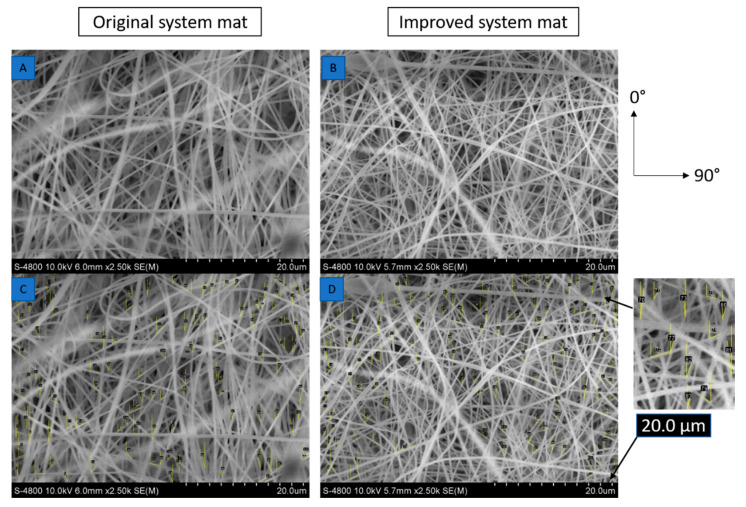
SEM images and angular measurements of mats from the (**A**) original and (**B**) improved systems. Orientations are measured using Image J software for (**C**) original and (**D**) improved systems. Additional data is available in the Appendix A.

**Figure 7 nanomaterials-10-02273-f007:**
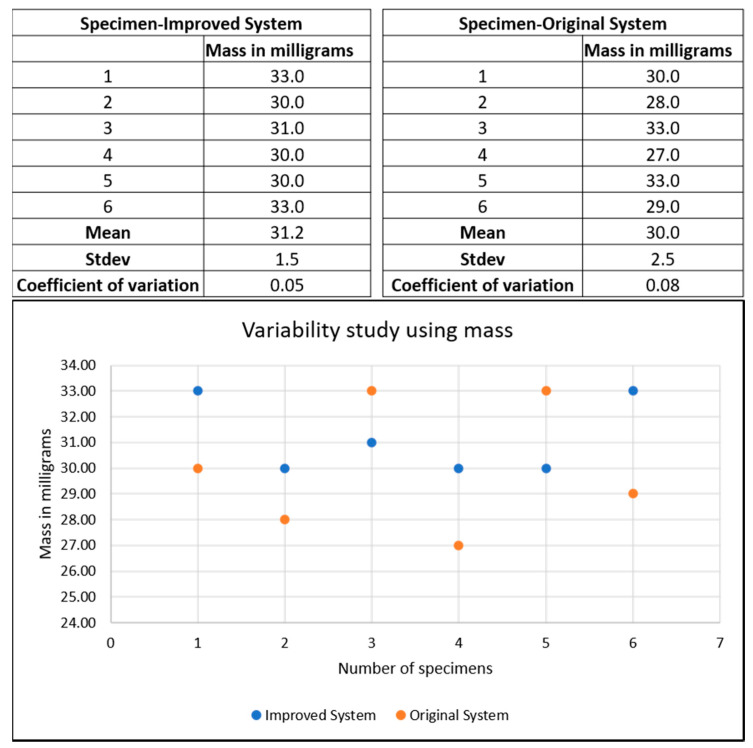
The improved system shows consistent mass distribution along the mat.

**Figure 8 nanomaterials-10-02273-f008:**
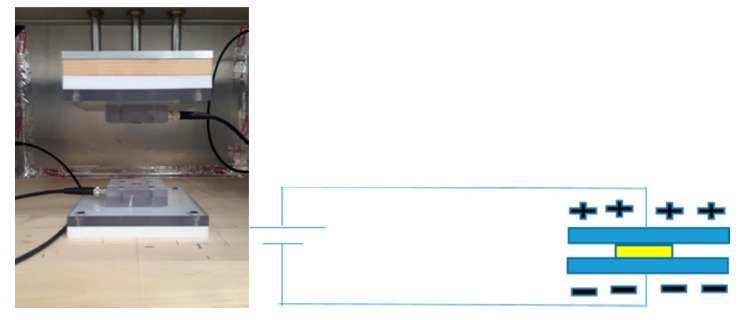
Test configuration and schematic representation of dielectric constant measurement.

**Figure 9 nanomaterials-10-02273-f009:**
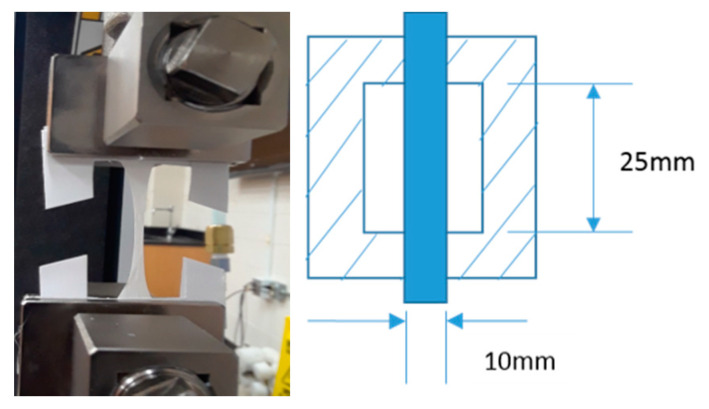
Test configuration and schematic of tensile test specimen.

**Figure 10 nanomaterials-10-02273-f010:**
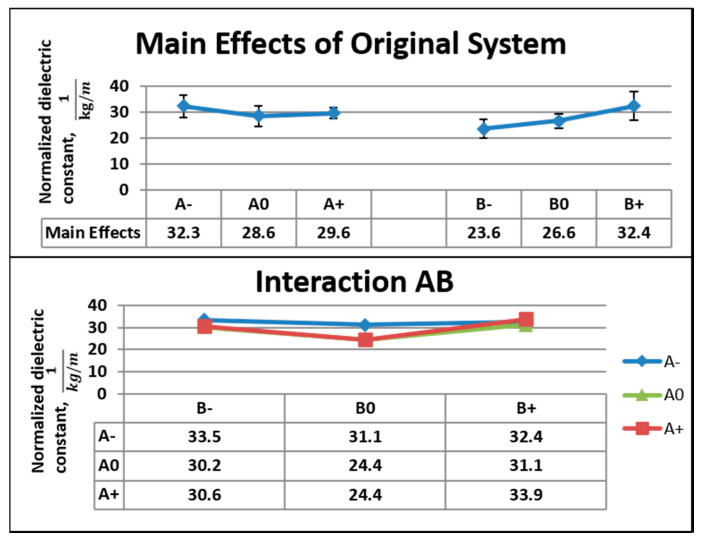
Main effects, standard deviation, and interactions for the dielectric data of the original system.

**Figure 11 nanomaterials-10-02273-f011:**
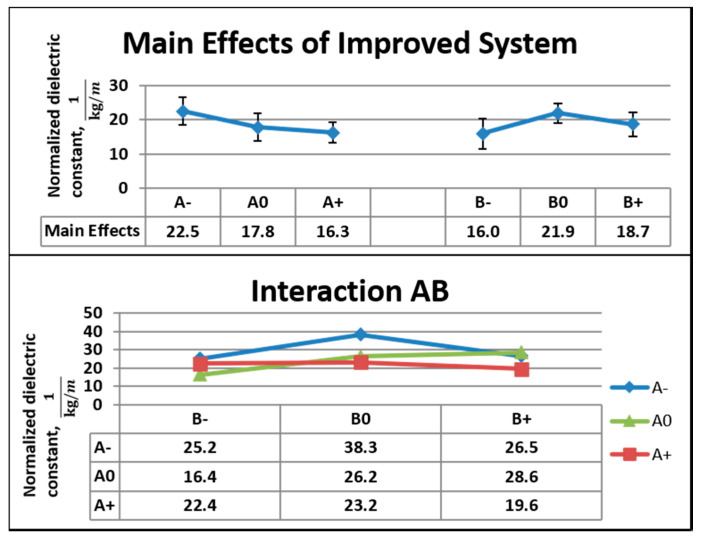
Main effects, standard deviation, and interactions for the dielectric data of the improved system.

**Figure 12 nanomaterials-10-02273-f012:**
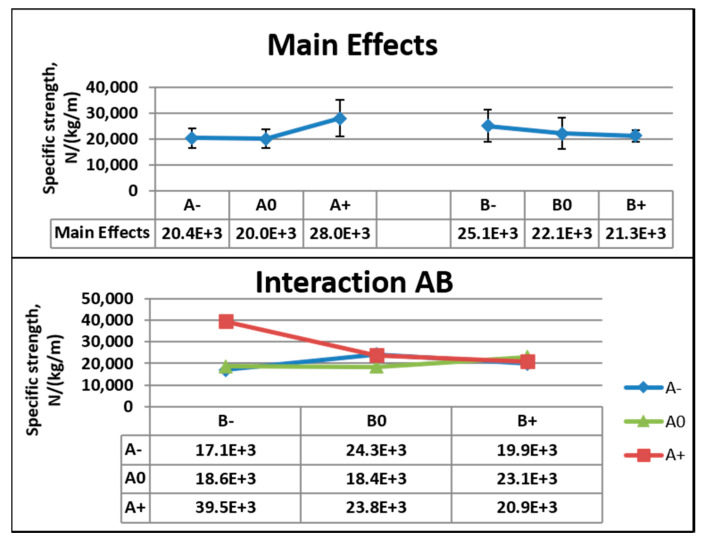
Main effects, standard deviation, and interactions for the tensile data of the original system.

**Figure 13 nanomaterials-10-02273-f013:**
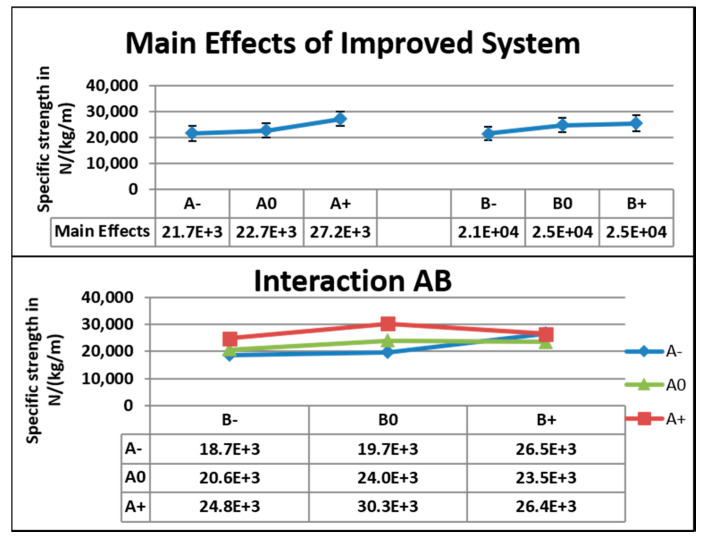
Main effects, standard deviation, and interactions for the tensile data of the improved system.

**Figure 14 nanomaterials-10-02273-f014:**
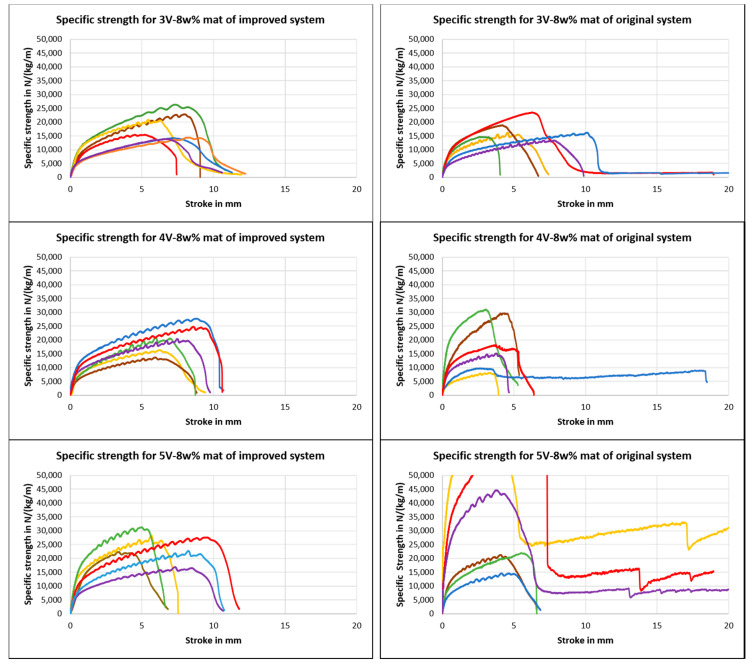
Tensile strengths of nanofiber mats of the improved (**left**) and original systems (**right**). Green, yellow, and brown colors indicate specimens from same mat, and red, blue, and violet colors indicate specimens from another mat of the same configuration.

**Figure 15 nanomaterials-10-02273-f015:**
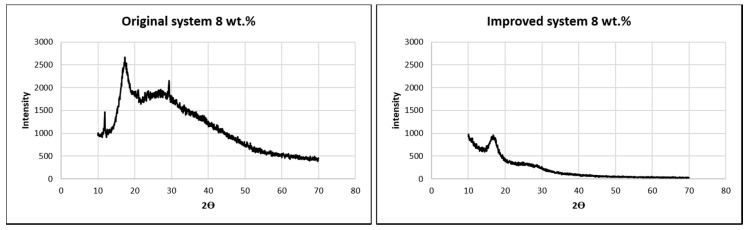
Intensity vs. 2ϴ plot of XRD analysis of PAN at 8 wt.% for the original and improved systems.

**Figure 16 nanomaterials-10-02273-f016:**
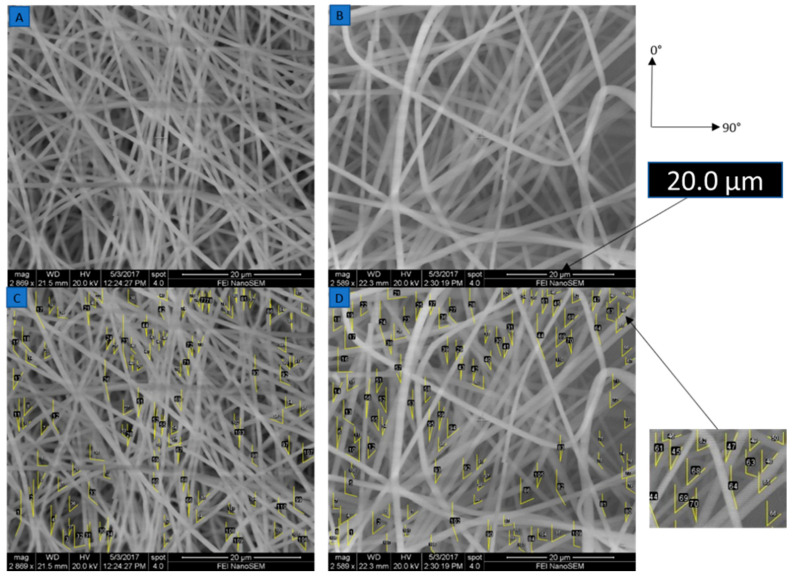
SEM images and angular measurements of mats from the (**A**) 4 V–8 wt.% mat, and (**B**) 5V–8 wt.% mat. Orientations are measured using Image J software for (**C**) 4 V–8 wt.% mat and (**D**) 5 V–8 wt.% mat. Additional data is available in the Appendix A.

**Figure 17 nanomaterials-10-02273-f017:**
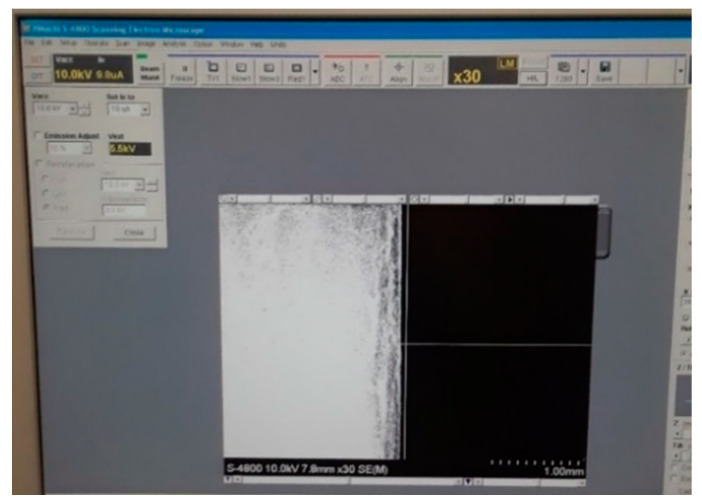
The fiber direction side of mat is in coincidence with the vertical axis of the SEM device.

**Figure 18 nanomaterials-10-02273-f018:**
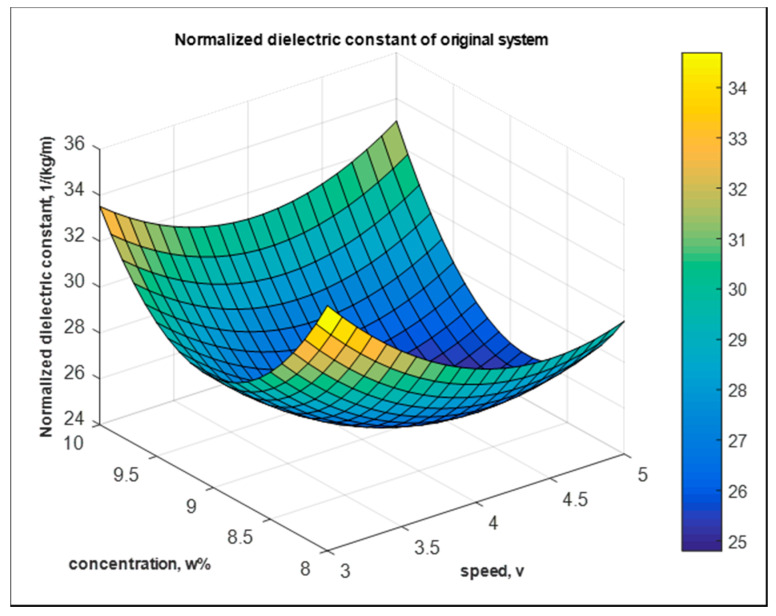
Fitted model of normalized dielectric constant from the original system.

**Figure 19 nanomaterials-10-02273-f019:**
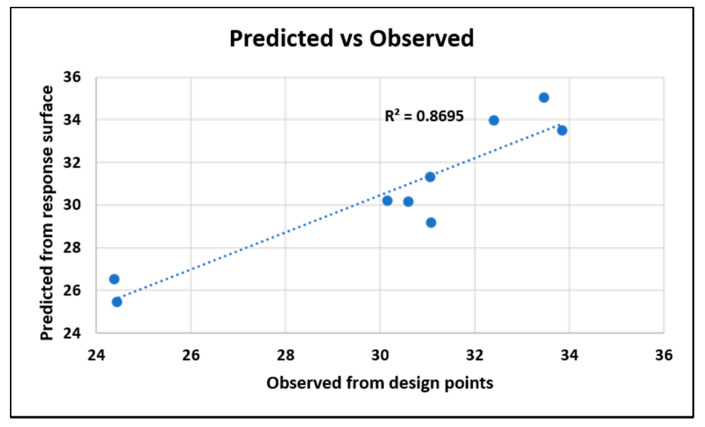
Goodness of fit for the original system from the dielectric response.

**Figure 20 nanomaterials-10-02273-f020:**
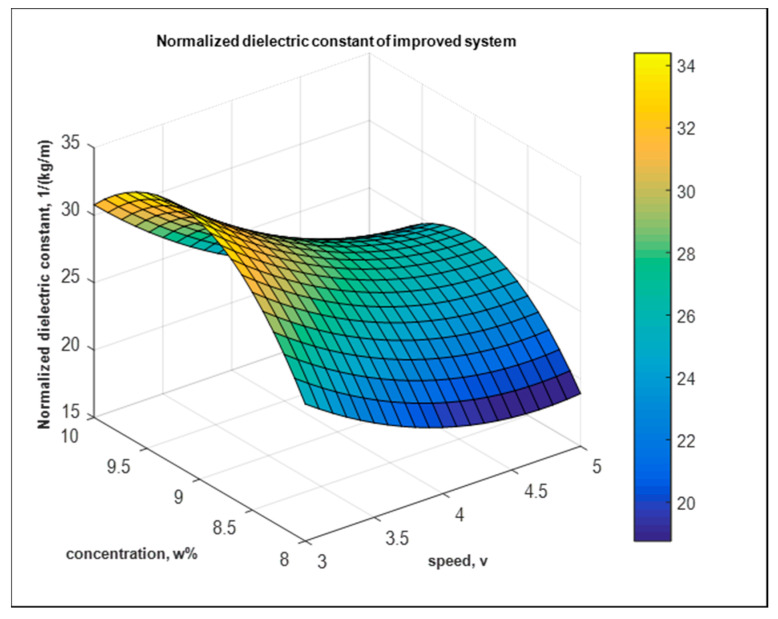
Fitted model of normalized dielectric constant of the improved system.

**Figure 21 nanomaterials-10-02273-f021:**
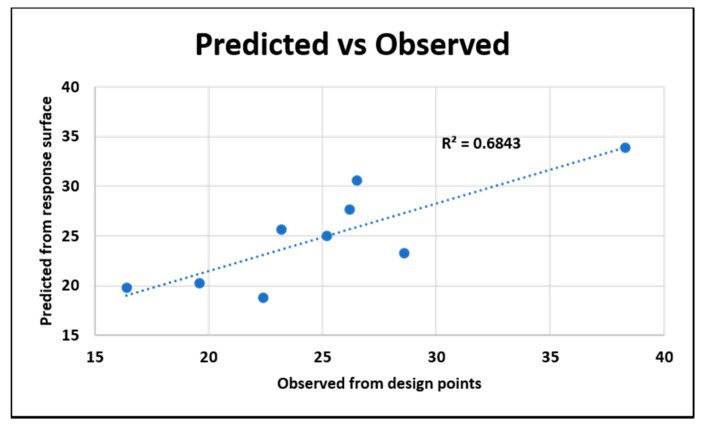
Goodness of fit for the improved system from the dielectric response.

**Figure 22 nanomaterials-10-02273-f022:**
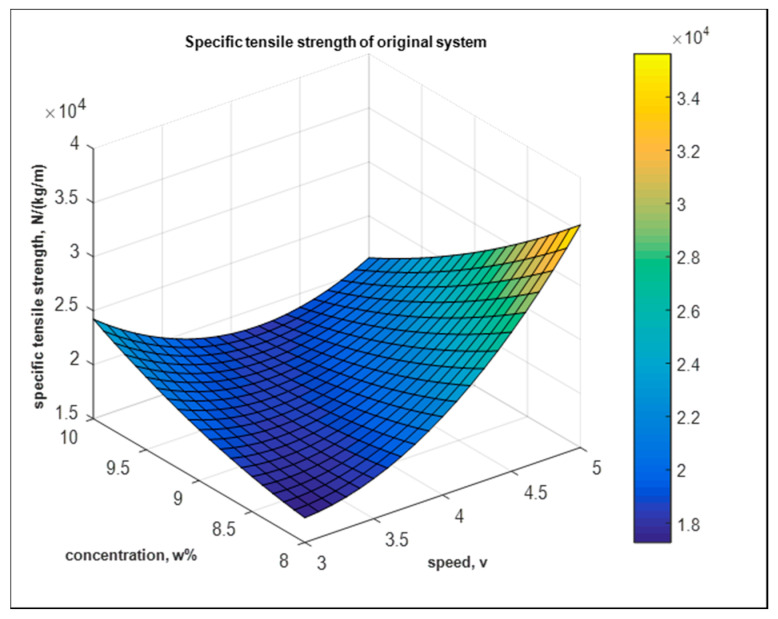
Fitted model of the specific tensile strength from the original system.

**Figure 23 nanomaterials-10-02273-f023:**
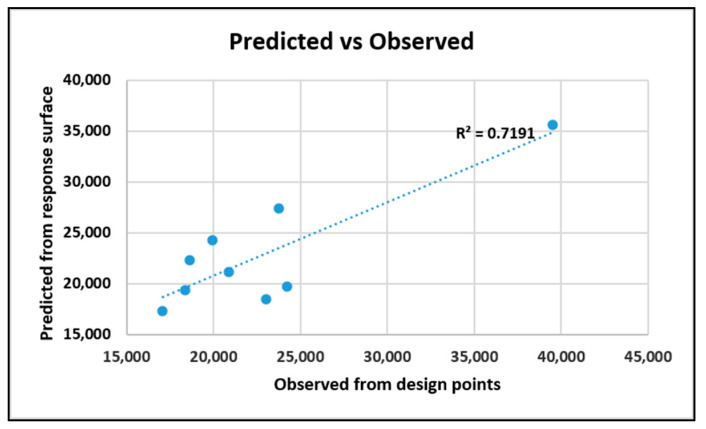
Goodness of fit for the original system from tensile response.

**Figure 24 nanomaterials-10-02273-f024:**
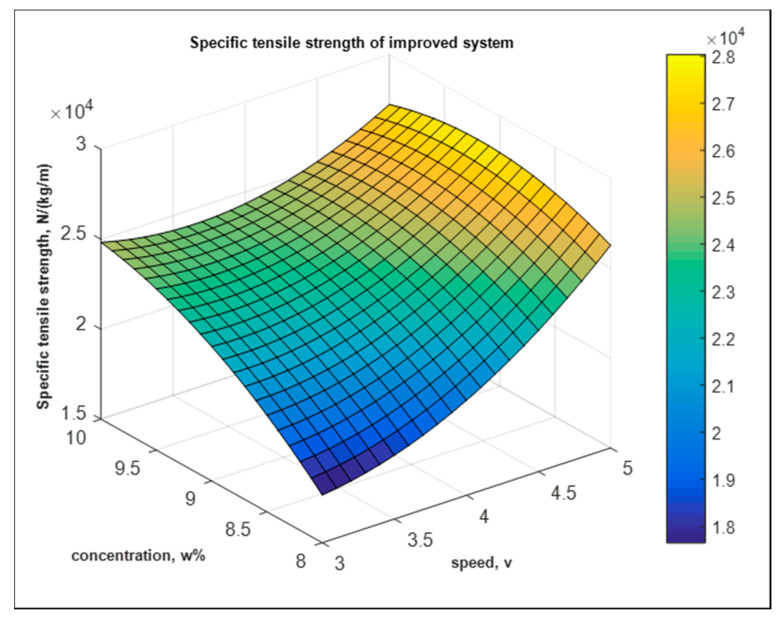
Fitted model of specific tensile strength from improved system.

**Figure 25 nanomaterials-10-02273-f025:**
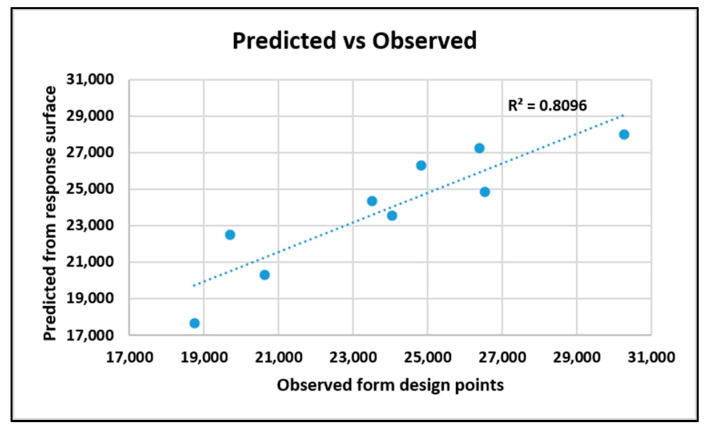
Goodness of fit for improved system from tensile response.

**Figure 26 nanomaterials-10-02273-f026:**
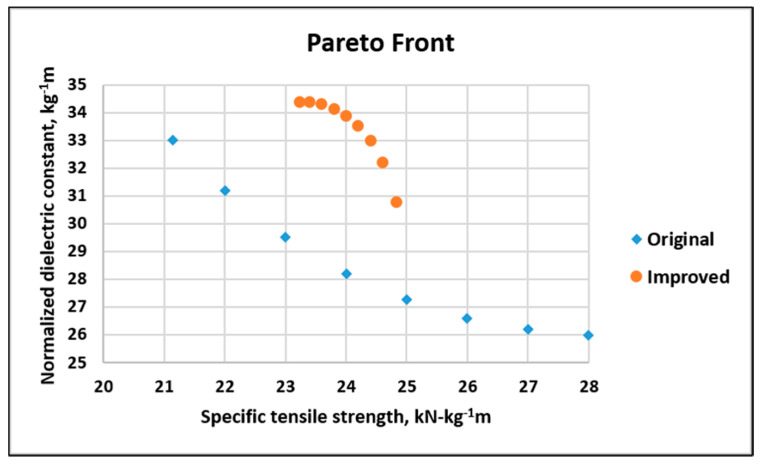
Dielectric versus tensile for the original and improved systems.

**Table 1 nanomaterials-10-02273-t001:** Test matrix for the original and improved systems.

	Speed A	Concentration B	Tensile	Dielectric
1	+	+	6	6
2	0	+	6	6
3	-	+	6	6
4	+	0	6	6
5	0	0	6	6
6	-	0	6	6
7	+	-	6	6
8	0	-	6	6
9	-	-	6	6
10			54	54

**Table 2 nanomaterials-10-02273-t002:** Two factors and three levels for each factor

	Speed A	Concentration B
**+**	5 V	10 wt.%
**-**	3 V	8 wt.%
**0**	4 V	9 wt.%

**Table 3 nanomaterials-10-02273-t003:** Improvement in specific tensile strength in improved system.

	Increase in Specific Tensile Strength				Percentage Increase		
	10 wt.%	9 wt.%	8 wt.%		10 wt.%	9 wt.%	8 wt.%
**5 V**	55.0 × 10^2^	65.2 × 10^2^	−14.7 × 10^3^	**5 V**	26.3%	27.5%	−37.1%
**4 V**	4.5 × 10^2^	56.4 × 10^2^	20.1 × 10^2^	**4 V**	1.9%	30.7%	10.8%
**3 V**	66.0 × 10^2^	−45.6 × 10^2^	16.6 × 10^2^	**3 V**	33.1%	−18.8%	9.7%

**Table 4 nanomaterials-10-02273-t004:** Improvement in dielectric constants in improved system.

	Increase in Dielectric Data				Percentage Increase		
	10 wt.%	9 wt.%	8 wt.%		10 wt.%	9 wt.%	8 wt.%
**5 V**	−14.3	−1.2	−8.2	**5 V**	−42.2%	−4.9%	−26.8%
**4 V**	−2.5	1.8	−13.8	**4 V**	−8.0%	7.4%	−45.7%
**3 V**	−5.9	7.2	−8.3	**3 V**	−18.2%	23.2%	−24.8%

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
