# Peer review of "Anisotropic Characterizations of Electrospun PAN Nanofiber Mats Using Design of Experiments"

_nanomaterials, 2020, doi:10.3390/nano10112273_

Round 1

Reviewer 1 Report

  1. Scale bar of Fig. 6 is not visible. It can be improved
  2. How with the different percentage of PAN solution the dielectric and mechanical property of nanofiber change need to be explained in abstract or conclusion
  3. There is no mentioned of linear regression model
  4. There is no clarity of fig. 16

Author Response

Dear Reviewer,

Thanks

Reviewer 2 Report

Dear Authors,

in your manuscript, the following points should be added/changed to further improve it:

  • line 29: You mention a voltage per distance before you even mention electrospinning, so this needs some more explanation. Besides, it is necessary to mention that this value is only valid for needle-electrospinning; thinking about the wire-based technique, PAN droplets on the wire wouldn't even start moving, not to talk about spinning.
  • line 53: Please explain the beta-phase increment
  • line 63: "8 wt% of solute and solvent ... this sounds strange, probably you mean "of solute in the solvent"
  • viscosity: In which way did you measure it?
  • Needle: The dimensions are 0.7 mm x 40 mm (one unit is missing).
  • line 69: voltages with capital V, please
  • line 86: Please add the units to the calculation (23 kV/(20 cm))
  • Although you present three sketches of the improved system, I still don't understand where the mesh enters the game. In line 92 you write "... form fibers on the mandrel", which is also shown in Fig. 2. But where is the mesh which is visible in Fig. 4, and why do the fiber leave the mandrel after impinging on it?
  • Fig. 6: Please add the usual information about the SEM used here. Why do you think that a standard deviation of 21° (standard deviations don't have more than two digits) is significantly larger than one of 19°? Besides, I am wondering about the angles given for the left image. There are fibers from upper left to lower right, and fibers from lower left to upper right, so there should be a difference of up to 90° (or more, depending on how you perform the calculation) in the angles. Where is the 0° angle? Please check these values.
  • Fig. 7: The lines between the dots are not allowed here since there are no values between the separate x-axis labels (there is no sample 2.3 or the like).
  • Table 2: Please add the units.
  • line 142: 0.5 inch x 0.5 inch
  • line 144: what is meant with "after" 1 kHz, above or below?
  • line 165: Why is G chosen instead of the usual m? G looks like the Gravity constant. Besides, rho from the formula is not identical with the density in line 165.
  • line 171: 2 inch x 1/2 inch
  • line 179: Now you have defined a parameter for the specific strength, so this should be added to the formula.
  • line 185: The porosity was defined before as a percentage value, so it cannot be inserted here.
  • line 193: "8.89" - which unit?
  • Fig. 10: Why is the standard deviation not just given in the main effect plot? This would make it much easier to see whether the apparent effects are significant or not. (ditto for Figs. 11 ff.) Corresponding to the relatively large standard deviations, too many digits of the values in the text should be avoided, such as 3772.8 (line 219) or even six-digit numbers below.
  • Line 246-247: the units don't need the "minus"
  • Fig. 14: The lines are hard to see, and so are the axis labels. it would be good to increase letter sizes and to finish the x-axis maybe around 21 or 22 mm to make everything better visible. Besides, we need explanations for the colors.
  • Table 3: Here, again, the standard deviations are identical, so why should one image show a better alignment? Besides, if you really want to evaluate the alignment and have so large standard deviations, then you really need much more fibers (usually we show values for min. 100 fibers from larger areas to avoid problems due to "cherry-picking" of microscopic images).
  • line 309: please be careful with the units. A and B have units, the dielectric constant not. What is e2? (ditto for the other fitting equations)
  • Table 4: Too many digits.
  • Fig. 26: How are the Pareto fronts calculated?
  • References: There is one from 2017, nothing from recent year. Please check the recent literature dealing with electrospinning PAN, spinning parameter optimization, fiber alignment methods etc.
  •  

Author Response

Dear Reviewer,

Thanks

Round 2

Reviewer 1 Report

This manuscript can be accepted in the present form.

Author Response

Dear reviewer,

Thanks for the comments.

I look forward to have this manuscript published.

Thanks

Reviewer 2 Report

Dear Authors,

after taking into account most of my suggestions, there are only few points left:

  • Fig. 6: In the left image, there are a lot of fibers running from upper left to lower right, so these fibers must have angles either below 0° or above 90°, depending on how you count. On the other hand, the mean values don't matter at all. All fibers can be fully aligned for a mean value of 67°, or they can be completely arbitrarily orientated for a mean value of 0°; this value only depends on where you decide to locate your coordinate system. The interesting value is the standard deviation - and this is nearly identical for both images. Besides, counting only 10 fibers is really insufficient to tell anything about the angular distribution.
  • Fig. 10: What I meant is: Please show the standard deviations as usually in the form of error bars. In the lower part of this figure, there are too many digits; it is just 32 +- 9 etc. (ditto in the text)
  • Fig. 14: Please add (just as insets) the meaning of the different colors in the sub-graphs.
  • Table 3: Please reduce the numbers of significant digits at least for the mean values. Besides, standard deviations of 42° and 43° are really not different, so again there is no difference in the alignment between these two nanofiber mats.
  • Equation for Y1: It is definitely not possible to use "e" instead of "exp"; "e" is always Euler's number which is not 10. So please write this properly; either (if necessary due to any unknown reason) by exp(2) or in the normal scientific style as 10^2. Besides, the units are still not correct. When you are adding different terms, then all these terms must have equal units. Here, the first term has no unit, while the others do have units. Normally this problem is simply solved by dividing each parameter by its unit.

Author Response

Dear reviewer,

Thanks
